# Glial Modulation of Energy Balance: The Dorsal Vagal Complex Is No Exception

**DOI:** 10.3390/ijms23020960

**Published:** 2022-01-16

**Authors:** Jean-Denis Troadec, Stéphanie Gaigé, Manon Barbot, Bruno Lebrun, Rym Barbouche, Anne Abysique

**Affiliations:** Laboratoire de Neurosciences Cognitives, Campus St Charles, Université Aix Marseille, UMR CNRS 7291, 13331 Marseille, France; stephanie.rami@univ-amu.fr (S.G.); manon.barbot@univ-amu.fr (M.B.); bruno.lebrun@univ-amu.fr (B.L.); rim.barbouche@univ-amu.fr (R.B.)

**Keywords:** metabolism, food intake, obesity, NTS, area postrema, astrocytes, microglia, vagliocytes, oligodendrocytes, leptin, glucose

## Abstract

The avoidance of being overweight or obese is a daily challenge for a growing number of people. The growing proportion of people suffering from a nutritional imbalance in many parts of the world exemplifies this challenge and emphasizes the need for a better understanding of the mechanisms that regulate nutritional balance. Until recently, research on the central regulation of food intake primarily focused on neuronal signaling, with little attention paid to the role of glial cells. Over the last few decades, our understanding of glial cells has changed dramatically. These cells are increasingly regarded as important neuronal partners, contributing not just to cerebral homeostasis, but also to cerebral signaling. Our understanding of the central regulation of energy balance is part of this (r)evolution. Evidence is accumulating that glial cells play a dynamic role in the modulation of energy balance. In the present review, we summarize recent data indicating that the multifaceted glial compartment of the brainstem dorsal vagal complex (DVC) should be considered in research aimed at identifying feeding-related processes operating at this level.

## 1. Introduction

Maintaining a healthy weight has turned into a personal difficulty as well as a serious public health concern. In fact, the majority of the world’s population lives in countries where being overweight or obese kills more people than being underweight, indicating the growing number of people who suffer from a nutritional imbalance. Since 1975, the global obesity rate has nearly tripled. Over 1.9 billion adults (aged 18 and over) were obese in 2016. A further 650 million were overweight. According to the World Health Organization (WHO) [1], 39 million children under the age of five were projected to be overweight or obese in 2020, with over 340 million children and adolescents aged 5–19 also being overweight or obese. Despite the tremendous amount of information available, these alarming findings highlight the need for a deeper understanding of the mechanisms that control energy balance. Decoding more precisely the mechanisms regulating energy balance remains a necessary challenge if we hope to effectively treat metabolic disorders. Body weight control appears complex and multifaceted, and involves many factors controlling both food intake and energy expenditure in the metabolic balance. By controlling hunger and glucose homeostasis, the brain plays an essential role in regulating food intake and energy expenditure. Two central regions, i.e., the hypothalamus and DVC, a brainstem structure, strongly contribute to the homeostatic control of energy balance by integrating information linked to nutritional status and arising from peripheral organs (the gut, liver, pancreas, and adipose tissue; see [2] for review). In these structures, neuronal networks dedicated to the regulation of energy balance have been extensively studied [3]. More recently, many studies have strongly suggested that neuroglial interactions contribute to the regulation of feeding behavior and control of energy balance (see [4,5] for reviews). Studies demonstrating that neuroglial interactions within the hypothalamus finely tune energy homeostasis are now relatively abundant and were recently reviewed in [4]. Evidence that glial cells could also modulate energy balance at the brainstem level has emerged more recently, even though we have advanced this idea for a decade [6]. Interestingly, in recent years, exciting papers illustrating both the diversity of glial populations within the DVC and the presence of neuroglial interactions have been published. Thus, we here propose to gather the data now available suggesting that the DVC glial populations contribute to the modulation of energy balance. We focus our review on energy balance, although we are aware that at the brainstem level, neuroglial interactions interfere with other autonomic functions, particularly cardiorespiratory function. Nonetheless, this field has already been discussed in recent reviews [5,7]. Here, we describe the structural organization and diversity of DVC glial cells and their contributions to: (i) glucodetection and associated responses; (ii) the integration of viscerosensory signals; and (iii) the modulation of food intake.

## 2. The Adult DVC: Structural and Functional Aspects

The regulation of energy balance requires the complex integration of homeostatic signals arising from the periphery via neuronal and humoral pathways. In this context, the DVC, a brainstem structure that mainly contains the nucleus of the solitary tract (NTS), area postrema (AP), and dorsal motor nucleus of the vagus nerve (DMNX), integrates a large volume of gastrointestinal sensory afferent information and participates in the coordination of efferent responses to regulate food and caloric intake, pancreatic exocrine secretion, and gastric/intestinal motility and emptying (Figure 1). Most of the gastrointestinal sensory information is conveyed by the vagal afferents and projects to the NTS and, to a lesser extent, to the AP. The vagus nerve (X cranial nerve) is the key neuroanatomical link between the gastrointestinal tract and the brain. These vagal afferents are critical for the regulation of food intake, since surgical vagal deafferentation results in the consumption of larger meals than those consumed by sham-operated controls [8]. The NTS is the main relay of visceral afferents and contains viscerotropic representations of the alimentary tract [9]. For instance, sensory inputs related to taste from the tongue project to the ventral NTS, whereas afferents from the gastrointestinal tract project to the intermediate and caudal NTS [10]. The NTS integrates sensory inputs from the gastrointestinal tract, as well as the cardiothoracic and respiratory systems [9]. Furthermore, the NTS projects to and receives input from other midbrain and forebrain nuclei, including the parabrachial nucleus (PBN), hypothalamus nuclei, and ventral tegmental area/substantia nigra (VTA/SN; see Figure 1). This NTS connectivity with other central nervous system (CNS) regions implies that these areas can modulate gastrointestinal vagal efferent outputs; it also allows the NTS to influence eating behavior. Neurons in the NTS principally use glutamate, gamma-aminobutyric acid (GABA), and norepinephrine (NE), but it should be noted that many neuropeptides linked to the regulation of food intake, including glucagon-like peptide 1 (GLP-1), cholecystokinin (CCK), neuropeptide Y (NPY), and pro-opiomelanocortin (POMC), are expressed by NTS neurons [11]. Gastrointestinal sensory information is mainly relayed by glutamatergic terminals from vagal afferents and the subsequent activation of ionotropic glutamate receptors on NTS neurons [12]. Fourth-ventricle or NTS injection of the noncompetitive *N*-methyl-D-aspartate (NMDA) receptor antagonist MK-801 blunted the CCK-induced reduction of feeding produced by intraperitoneal (ip) CCK injection [13]. In addition to vagal afferents, NTS neurons receive hormonal and metabolic circulating information linked to adiposity, glycemia, and nutrient availability thanks to their close proximity to the AP, a circumventricular organ with fenestrated capillaries and a leaky blood–brain barrier (BBB). Peripheral CCK, for example, has been shown to activate NTS neurons directly following a vagotomy [14]. Thus, the metabolic signals released by the gastrointestinal tract can modulate the activity of NTS neurons both indirectly, via peripheral vagal afferent modulation, and directly, at the central NTS sites. These two modes of communication expand the potential for temporal and spatial modulation of NTS activity, allowing more precise control of feeding behavior [15]. The integrated information is both relayed to the top upper centers located in the midbrain and forebrain (see above) and to the DMNX to develop an appropriate response at the digestive tract level. Located ventral to the NTS, the DMNX contains the preganglionic parasympathetic motoneurons that provide efferent innervation to the esophagus, stomach, small intestine, and proximal colon [9]. The subdiaphragmatic branches of the vagus nerve feature a medial-lateral columnar organization within the DMNX [9]. DMNX preganglionic motoneurons are cholinergic and support both excitatory and inhibitory pathways to exert control over gastrointestinal functions, such as gastric emptying, intestinal transit, or absorption. These functions in turn influence feeding behavior.

As mentioned above, one of the notable DVC characteristics is the presence of AP, a circumventricular organ (CVO). The AP is located in the fourth ventricle on the dorsal surface of the medulla, adjacent to the NTS, and exhibits a rich vasculature of fenestrated capillaries. Interestingly, AP neurons send dendritic projections to vascular endothelial cells, allowing a higher exposure to circulating factors than other brainstem regions involved in controlling feeding behavior [16]. AP neurons express receptors for many neuromodulators, including CCK, GLP-1, ghrelin, and peptide YY (PYY). These receptors allow them to integrate metabolic information and participate not only in emetic response but, more broadly, in feeding behavior control. To support this claim, consider the studies that show that anorexigenic gut peptides such as CCK, GLP-1, and PYY modify AP neuron excitability (see [17] for review) and those that show that ghrelin-dependent stimulation of feeding requires an intact AP [18]. Although exposed to circulating signals, the AP also receives inputs from many regions involved in the regulation of feeding behavior, such as the NTS, PBN, and paraventricular nucleus (PVN). In return, AP sends projections to most of these centers. Thus, AP contributes with NTS to the integration of peripheral and central information linked to feeding control (Figure 1).

Finally, we can mention the *funiculus separens*, a part of DVC that is often neglected but that deserves to be highlighted. This region is an oblique ridge in the floor of the fourth ventricle, comprising a strip of ependymal cells separating the NTS and AP. This region contains atypical ependymocytes that resemble tanycytes of the third ventricle. Some evidence suggests that this zone regulates the diffusion of metabolically active circulating compounds from AP to NTS [19].

## 3. Cellular Diversity and Glial Organization within the Adult DVC

In the field of neuroglial interactions, astrocytes are salient due to their responsiveness to many neurotransmitters and the possible release of gliotransmitters (for a review, see [20]). It is therefore legitimate to focus on these cells as a priority when we want to identify neuroglial interactions within DVC. The simple detection of the emblematic marker of astrocytes, glial fibrillary acidic protein (GFAP), has revealed a striking pattern of GFAP immunoreactivity in DVC, which distinguishes it from the surrounding brainstem nuclei, i.e., the hypoglossal nucleus, dorsal tegmentum, and medullary reticular nucleus ([21,22,23] and Figure 2). While NTS and DMNX appear heavily labeled throughout their entire rostrocaudal extent, weak GFAP labelling is observed in AP [22]. In both NTS and DMNX, GFAP-immunoreactive cells exhibit the typical stellate morphology of differentiated protoplasmic astrocytes, but due to their strikingly high abundance compared to that of the surrounding brainstem nuclei, their possible neuromodulatory action was rapidly questioned [6], especially since several ultrastructural studies have illustrated the presence of perisynaptic astroglial processes [24,25]. An analysis of the postnatal development of NTS using glial glutamate transporters (GLAST, GLT-1) and GFAP as markers revealed that astrocytic processes are in the vicinity of small cell bodies, suggesting that astrocytes might be able to modulate axo-somatic synaptic transmission within NTS [24]. Chounlamountry and Kessler [25] demonstrated that the volume fraction of astrocyte processes and the astrocyte membrane density in adult NTS reached 15% and 2.8 m (-3), respectively, using GLT-1 immunodetection at the light and electron microscope levels. Interestingly, astrocyte processes contacted 58% of the perimeter of NTS single synapses, and glial coverage averaged 50% in multisynaptic arrangements [25]. Furthermore, changes in astrocyte coverage within the NTS and DMNX have been observed in a variety of models, including vagotomy [26], hypoxia [27], diesel exhaust particle exposure [28], and ozone inhalation [29]. Interestingly, NTS astrocytes were also shown to respond to changes in energy status [30]. Indeed, mice fed with a high-fat diet (HFD) for 12 h exhibited NTS astrocyte activation, as evidenced by an increase in the number (65%) and morphological complexity of GFAP+ cells, especially in NTS areas adjacent to AP [30]. 

In addition to protoplasmic astrocytes, which are in close contact with pre- and postsynaptic compartments, are actively engaged in synaptic development, and function in most parts of the mammalian brain, the DVC comprises an atypical glial cell population that morphologically resembles the specialized ependymal tanycytes cells bordering the ventral third ventricle. Indeed, these cells exhibit atypical astrocyte morphology with long, thin, and unbranched processes originating in cuboid-shaped GFAP+ somata localized in the ependyma of the fourth ventricle. These processes form a bundle at the AP/NTS interface, i.e., *funiculus separans* and midline commissural NTS ([22]; Figure 2). In 2007, we first described these cells and showed that they co-express nestin and vimentin in addition to GFAP. By analogy with the tanycytes of the third ventricle, we named these tanycyte-like cells “*vagliocytes*” in reference to the vagal complex and to differentiate them from the first cells [6]. Maolood and Meister [31] reported the immunoreactivity of DARPP32, a marker classically used to identify tanycytes, in the AP and at the borderline between AP and NTS, reinforcing the idea of a strong homology between glial radiating cells present at both the third and fourth ventricle borders. The morphological characteristics of these cells are evocative of developmental radial glia [32]. Besides, the immunohistochemistry (IHC) of GFAP, vimentin, and nestin from P0 to P30 has shown that vagliocytes are present in the AP/NTS borders and commissural NTS at birth, with only small changes in distribution and labelling density during early postnatal development [22]. A recent single-nucleus RNA-sequencing data analysis confirmed the presence of these tanycyte-like cells. Focusing their analysis on AP tissue harvested using anatomical landmarks to minimize incorporation of the adjacent NTS region, Zhang et al. [33] revealed that tanycyte-like cells with a specific genetic signature accounted for one third of all AP cells. Another single-nucleus RNA-sequencing study on the AP and adjacent NTS also confirmed the presence of a unique glial population in the AP that presented similarities to the arcuate nucleus (ARC) tanycytes [34], compared to a previously published ARC-median eminence (ME) single-cell atlas [35]. The morphology and position of these vagliocytes have quickly led to the hypothesis that they could constitute a barrier to the diffusion of elements, having joined the AP through fenestrated capillaries present in this CVO [6,19]. A hypothesis has been formulated for tanycytes lining the ME and ARC [36,37,38]. In a pioneering work, Maness et al. reported the lack of diffusion of intravenously (iv) injected tracers out of the CVOs, contradicting the widespread idea that CVOs constitute an open window allowing diffusion of circulating molecules from blood to surrounding parenchyma [39]. It was also shown that the iv injection of 0.3 kDa sodium fluorescein results in the rapid labelling of both the AP and the surrounding NTS, while the diffusion of fluorescent labelled dextran >3 kDa is restricted to the AP following its systemic injection [40]. Similarly, it was observed that the fluorescent dye hydroxystilbamidine (OHSt, 0.47 kDa) is largely confined to the AP after iv administration. Finally, the AP/NTS border has been shown to be immuno-positive for zonula occludin-1 (ZO-1), a protein constitutive of tight junctions [31,41]. The glial identity of ZO-1 positive cells was established by GFAP immunostaining coupled to electronic microscopy [41]. Some years ago, we reported the in vivo and in vitro expressions of leptin receptor (LepR) isoforms at the AP/NTS glial boundary level, with a strong expression of short isoforms. These isoforms may play a role in receptor-mediated leptin transport from the AP to the NTS [19]. More recently, vagliocytes were reported to express octadecaneuropeptide (ODN), a glial anorexigenic peptide [42].

Microglia cells are considered the “immune cells of the brain” and play key roles in regulating brain development and homeostasis, as well as in repairing injuries. Microglia are found throughout the brain and spinal cord [45,46], accounting for 5–20% of the total glial cell population within the CNS parenchyma. Currently, the microglia structure and function in the DVC remain poorly characterized. Using CD11b-immunostaining, a surface marker specific to reactive microglia, Maolood and Meister [31] were the first to report rich microglia coverage in AP, NTS, and DMNX, with a preferential association with vessels. More recently, it was reported that a microglial subpopulation characterized by amoeboid active shape and expression of both M1 markers, such as CD16/32, CD86, and M2 markers, such as CD206 and Ym1, is present in CVOs, including AP. Activated under normal conditions, this microglia population could serve as a sentinel in CVOs with regard to the entry of blood-derived molecules through fenestrated capillaries [43]. Interestingly, the DVC microglial population has been shown to respond to stimuli and lesions. Vagotomy increases NTS microglial cell numbers and produces morphological changes indicative of microglia activation [26]. Furthermore, microgliosis, comprising an increase in the number of microglial cells and a decrease in microglial branching, was observed in the NTS after short-term HFD consumption (4 days; [44]). Vaughn et al. [47] also reported an increase in microglia activation in the NTS and DMNX following a longer period of HFD consumption. Furthermore, the inhibition of microglial activation by minocycline, an inhibitor of microglia, reduced body fat accumulation [47]. Altogether, these data suggest that microglia should be considered in future studies aiming to understand the role of DVC in the normal and pathological regulation of energy balance.

## 4. DVC Glial Cells, Glucodetection and Regulatory Responses

Glucose homeostasis is of critical importance to human health since glucose constitutes the primary source of energy. As a result, a sufficient glucose level in the blood is necessary for survival. On the other hand, hyperglycemia is a primary symptom of diabetes. This balance is maintained through a fine interplay between central and peripheral mechanisms, allowing the storage of excess glucose after meals and its mobilization from stores during periods of fasting. At the central level, glucosensing cells that integrate information regarding glucose availability are distributed in the brainstem, pontic structures, and hypothalamus [48]. The NTS has long been linked to the central detection of glucose availability and glucose homeostasis control. Indeed, NTS cells enjoy unique access to serum glucose levels due to their close proximity to AP [49]. A subgroup of NTS neurons, mainly tyrosine hydroxylase (TH)-positive catecholaminergic neurons, has been shown to respond to physiological changes in extracellular glucose concentrations both in vivo [50,51] and in vitro [52,53]. In response to hypoglycemia, a subpopulation of NTS TH neurons elicits a counter-regulatory response that includes glucagon and corticosterone secretion and increases food intake and sympathetic tone [54]. In this context, the systemic administration of the glucoprivic compound 2-deoxy-glucose (2-DG), a glucose analog commonly used to provoke the counter-regulatory response, has been reported to result in increased c-Fos expression in NTS and AP [51,55]. Another glucoprivic agent, 5-thio-D-glucose, was administered into the NTS and also led to an increase in food intake [56], as well as glucagon [57] and stress hormone [58] secretion. It was recently reported that TH-expressing neurons activated by glucoprivic stimulus project to the hypothalamus and elicit feeding through the bidirectional adrenergic modulation of agouti-related peptide (AgRP)- and POMC-expressing neurons [59]. By contrast, there is undoubtedly another subpopulation of TH+ NTS neurons that is inhibited by hypoglycemia, since some NTS TH neurons have been shown to respond to glucose concentration decreases by reducing their potential firing. Their firing rate increases when glucose returns to its resting level ([60]; Figure 3).

Initially, a small number of studies suggested that NTS glia may be involved in mechanisms of glucodetection, but this idea has gradually gained ground as data have accumulated. NTS astrocytes are now thought to be the primary glucosensors, activating hindbrain TH neurons, which in turn drive counter-regulatory response during glucose deficiency [61,62]. In a pioneering study, Young et al. [63] reported that methionine sulfoximine (MS), a drug that specifically impedes astroglial metabolism, significantly reduced the response of NTS neurons to the peripheral injection of the glucoprivic agent 2-DG [63]. This work was one of the first to advance the idea that NTS glial cells can act as glucose sensors. More recently, it was reported that the administration of fluorocitrate (FC; a blocker of astrocytic metabolism) in the fourth ventricle suppresses glucoprivation-induced increases in blood glucose levels [64]. Furthermore, NTS astrocytes appear to regulate the vago–vagal reflex that mediates increases in gastric motility during low glucose availability. Indeed, FC applied to the fourth ventricle blocks the increase in gastric motility provoked by either central or peripheral 2-DG-induced glucopenia [65]. Using brainstem slices, McDougal et al. [66,67] reported that the bath application of a solution containing either 0.5 to 2.5 mM glucose or glucoprivic 2-DG resulted in an intracellular calcium increase in both NTS-astrocytes and neurons. They elegantly showed that the blockage of communication between neurons and astrocytes with tetrodotoxin did not alter astrocytic responsiveness to low glucose concentration [66,67]. Moreover, using brainstem slices from TH-GCaMP5 transgenic mice, in which the calcium reporter construct (GCaMP5) was expressed selectively in tyrosine hydroxylase neurons, Rogers et al. [68] showed that low glucose and 2DG challenges increased intracellular calcium in approximately 90% of TH-GCaMP5 NTS neurons. This effect was blocked by FC and was mimicked by ATP in the presence or absence of FC co-treatment. Interestingly, the glycaemia increase induced by the application of 2-DG was blocked by caffeine, a non-selective adenosine antagonist, or by 8-Cyclopentyl-1,3-dipropylxanthine (DPCPX), a potent adenosine A1 antagonist [64]. Similarly, systemic hyperglycemia induced by thrombin injection into the hindbrain was blunted by FC or adenosine receptor antagonists (caffeine and DPCPX; [69]). Altogether, these results strongly suggest that NTS astrocytes detect glucopenia and, in turn, release adenosine as a gliotransmitter to induce the activation of downstream neuronal circuits responsible for the counter-regulatory response ([61,62]; Figure 3).

The glucose transporter-2 (GLUT-2), a low-affinity glucose transporter, is required for insulin release in response to hyperglycemia in pancreatic beta cells, which are archetype glucose-sensitive cells. Interestingly, GLUT-2 was shown to be expressed in a sub-population of astrocytes within the DVC [70]. Moreover, transgenic mice that only express GLUT-2 in pancreatic beta cells, i.e., whose central GLUT-2 expression was knocked out, exhibited defects in the control of glucagon secretion in response to low or high glucose [71]. Using a GLUT-2-null mouse model expressing a transgenic glucose transporter in their beta cells to rescue insulin secretion, Marty et al. [72] reported that the glial re-expression of GLUT-2 within the DVC of GLUT-2 null mice restored glucagon secretion and c-Fos labelling in the brainstem in response to glucoprivation. The same group showed that in GLUT-2 null mice, daily food intake was increased, and feeding initiation and termination following a fasting period were abnormal. To determine whether this abnormal feeding behavior arose from suppressed glucose sensing, they evaluated feeding in response to ip or intracerebroventricular (icv) glucose or 2-DG injections. They observed that in GLUT-2-null mice, feeding was no longer inhibited by glucose or activated by 2-DG injections [73]. Finally, it was recently proposed that GLUT-2 signals through increased Ca2+ r in glucose-sensitive NTS astrocytes. Indeed, co-immunoprecipitation assays revealed that GLUT-2 binds directly to the Gq protein subunit that activates phospholipase C (PLC). Moreover, using ex vivo calcium imaging performed on brainstem slices, it was confirmed that GLUT2 may be connected to a PLC-endoplasmic reticular-calcium release mechanism, since quercetin pretreatment, a GLUT-2 blocker, impeded 2-DG-induced calcium increase ([69]; Figure 3).

## 5. Integration of Viscerosensory Signals: Regulation by NTS Astrocytes

Vagal afferents enter the NTS via the solitary tract (ST), and the first step in the integration of the viscerosensory information they convey takes place in the gating of this input. A seminal paper involving NTS astrocytes in this process was published by McDougal aet al., who used horizontal brainstem slices previously loaded with a cell-permeant calcium indicator and fluorescent vital dye for astrocyte sulforhodamine 101 (SR101). The electrical stimulation of vagal afferents in the ST led to a rapid increase in intracellular calcium in astrocytes, involving calcium entry through a-amino-3-hydroxy-5-methyl-4-isoxa-zolepropionic acid receptors (AMPAR), which was amplified by calcium-induced calcium release from internal stores. This study also demonstrated that more than 70% of NTS astrocytes express AMPAR [74]. This proposed widespread use of fast ionotropic receptor-driven calcium signaling in NTS astrocytes contrasts sharply with the much slower metabotropic receptor-driven calcium responses in the hippocampus [75,76] and nucleus accumbens astrocytes [77] that were described previously. Given the tripartite synapse hypothesis, this particularity of NTS astrocytes opens up the possibility of time-locked gliotransmission to bursts of local vagal afferent inputs. Accorsi-Mendonsa et al. [78] performed patch-clamp recordings of NTS astrocytes and confirmed the presence of inward currents rapidly driven by ST stimulation and blocked by AMPAR antagonist. They also studied the effects of this astrocyte activation on ST-evoked and spontaneous excitatory postsynaptic currents (EPSC) recorded on identified NTS neurons sending projections to the ventral medulla. The inhibition of astrocytes through the bath application of the precursor of FC, fluoroacetate (FAC), reduced the amplitude of ST-evoked EPSC and spontaneous EPSC frequency on NTS neurons, both through a presynaptic mechanism. They elegantly identified ATP as the gliotransmitter, released by NTS astrocytes upon ST stimulation, which acts on the purinergic P2 receptor to stimulate presynaptic glutamate release on NTS neurons. It was shown that ST-induced astrocyte activation differentially modulates AMPA and NMDA currents in the postsynaptic NTS neurons, with an increase in AMPA currents (probably through the previously demonstrated presynaptic mechanism involving the gliotransmitter ATP) and a decrease in NMDA currents. Strikingly, the pre-exposure of rats to 24 h of sustained hypoxia did not change the amplitude of the ST-evoked AMPA currents in NTS astrocytes. By contrast, sustained hypoxia increased the amplitude of the ST-evoked AMPA and NMDA currents in NTS neurons and totally blunted the modulatory effects of the astrocyte inhibitor FAC [79]. In addition, it was shown that NTS astrocyte inhibition by FAC also modulates the intrinsic membrane properties of NTS neurons by inhibiting the A-type potassium current (IK_A_), a rapidly activating and transient K+ current. The inhibition of the IK_A_ current favors neuronal excitability and firing activity. The mechanism involved in this glial modulation of NTS neurons has not been studied so far [80]. Similar findings on astrocyte modulation of hypothalamic magnocellular neurosecretory neurons were previously studied in detail [81]. The revealed mechanism was that glutamate spillover upon glial excitatory amino acid transporter (EAAT) failure activates extrasynaptic NMDA receptors and triggers the inhibition of IK_A_ current in a calcium- and protein kinase C-dependent manner. Interestingly, alterations in extracellular glutamate buffering can result from astrocyte process retraction. This phenomenon may occur in several circumstances in the NTS owing to the high morphofunctional plasticity of NTS astroglial coverage (reviewed in [7,30]). The depolarization and action potential firing recorded in NTS neurons following pan-EAAT inhibition by threo-beta-benzyloxyaspartate (TBOA; [82]) or the selective inhibition of EAAT2 by dihydrohainate (DHK; [83,84]) can predict the consequences of reduced astroglial synapse ensheathment. Another effect of glial EAAT failure is the drying up of the glutamate/glutamine recycling process; its impact on vagal afferent input is attested by the reduction in ST-evoked EPSC in NTS neurons upon TBOA or DHK treatment.

Pellerin and Magistretti [85] proposed that the lactate shuttle provides a sufficient amount of the energy substrate, lactate, to sustain glutamatergic transmission. Sodium entry in astrocytes as a payload for glutamate capture stimulates Na/K ATPase and anaerobic glycolysis, leading to the production of lactate, which is supplied to neurons via complementary monocarboxylate transporters (MCT) present in astrocyte and neuronal membranes. With respect to vagal viscerosensory inputs, the pharmacological blockade of MCT in rat brainstem slices was shown to reduce ST-evoked EPSC on NTS neurons, an effect rescued by lactate bath application [86]. The gating of viscerosensory inputs also relies on inhibitory interneurons. Indeed, it was recently demonstrated that vagal afferents directly stimulate a high proportion of interneurons expressing somatostatin (SST) and releasing both GABA and glycine. These SST interneurons do not make contact with vagal afferents, but they do perform extensive feedforward inhibitory gating of viscerosensory signals [87]. Interestingly, astrocytes were recently shown to respond to SST interneurons with robust GABA_B_ receptor-mediated calcium elevations in the sensory cortex, an effect further strengthened in an SST-dependent manner upon repetitive SST interneuron firing [88]. Similarly, hippocampal astrocytes were shown to respond to SST interneurons and to enhance dendritic inhibition on pyramidal neurons through an adenosine/A1R mechanism [89]. Whether similar interactions between astrocytes and SST interneurons take place within the DVC remains to be explored.

## 6. Modulation of Food Intake and Weight Gain by DVC Glial Cells

As described above, DVC glial cells are involved in glucose sensing and play a key role in triggering the counter-regulatory response to glucoprivic challenge. In addition, DVC astrocytes play a modulatory role in the gating of the vagal afferents’ viscerosensory inputs through several mechanisms affecting glutamatergic transmission. These features obviously make them potential agents of food intake control. Interestingly, MacDonald et al. [30] recently reported that animals fed with HFD for 12 h exhibit a higher number of GFAP+ astrocytes and a greater number of processes per GFAP+ cell within the NTS compared with standard chow-fed controls. Although interesting, these observations do not constitute a demonstration of the causal involvement of DVC astrocytes in the modulation of feeding behavior. However, in the same study, the authors provided more direct evidence. Using a designer receptor exclusively activated by designer drugs (DREADD) strategy targeting DVC astrocytes, they proceeded to the specific chemogenetic activation of these cells and determined the impact on feeding behavior in mice. This chemogenetic activation of DVC astrocytes by clozapine *N*-oxide was found to decrease food intake and induce c-Fos expression in neighboring neurons [30]. This study is extremely interesting because it supports the idea that DVC astrocytes could modulate the energy balance. However, the underlying molecular mechanisms remain unknown. Several questions naturally arise from these observations: Which signaling pathways are able to modulate the activity of glial cells in the DVC? Which substances released by glial cells are capable of modifying the activity of the neuronal circuits involved in regulating food intake? By what means are they released? The answers to these questions are starting to become available, even though the data are still parceled (Figure 4).

GLP-1 is a short-half-life incretin hormone secreted by intestinal L cells. In addition to enhancing insulin release from pancreatic beta cells in a glucose-dependent manner, GLP-1 performs additional non-incretin functions, including glucagon secretion repression, gastric motility inhibition, and satiety enhancement [90]. GLP-1 acts through a single G protein-coupled receptor, GLP-1R, which is widely expressed in several peripheral organs as well as in the brain (for a review, see [91]). Several long-acting GLP-1R agonists have been approved for treating type 2 diabetes; two, liraglutide and semaglutide, have been approved as treatments for chronic weight management [92]. Recent studies using IHC with highly selective antibodies provided a complete atlas of GLP-1R in the mouse [93,94] and rat [95] brains. GLP-1R protein is found throughout the rostrocaudal extend of the rodent brain, with the highest levels in the CVO, including the AP and ME, as well as in the adjacent nuclei, ARC and NTS. Using whole-brain light sheet fluorescence microscopy in mice, Gabery et al. [94] compared a map of the brain areas accessible to a fluorescent semaglutide derivative after peripheral treatment to IHC maps of GLP-1R and semaglutide-induced c-Fos. Semaglutide label mapping was concordant with GLP-1R IHC, mainly in the hypothalamus and brainstem. By contrast, semaglutide-induced c-Fos only overlapped with the semaglutide label in AP, NTS, and DMNX and extended to other structures corresponding to lateral PBN and its projections [94]. The DVC therefore appears to be the primary GLP-1-sensitive brain area directly contacted by peripherally applied GLP-1R agonist to engage downstream anorectic circuitries. An interesting question is whether the DVC glial populations play a role in the anorectic effects of GLP-1R agonists. In 2016, Reiner et al. [96] showed that the fluorescent GLP-1R agonist fluoro-exendin 4, delivered in the fourth ventricle or intraperitoneally in rats, was captured by both NTS neurons and astrocytes. Moreover, icv pretreatment with the GLP-1R antagonist exendin 9 abolished the capture of intraperitoneally administered fluorescent exendin 4. Using calcium imaging in ex vivo brainstem slices, the authors showed that about 40% of NTS SR101-labelled astrocytes responded to exendin 4 with an increase in intracellular calcium, an effect significantly reduced by pre-treatment with exendin 9. Importantly, the inhibition of astrocytes by FC injected into the medial NTS immediately before exendin 4 injection abolished the anorectic effects of the GLP-1R agonist. Altogether, these results show that NTS astrocytes play a causal role in the anorexic effects induced by exendin 4 and suggest that a subpopulation of NTS astrocytes could directly respond to GLP-1R agonists. More recently, in mice treated with semaglutide, single-nucleus (sn)RNA sequencing showed massive transcriptomic modifications in DVC astrocytes and tanycyte-like cells [34,97]. We cannot exclude that these effects are indirectly induced by the activation of GLP-1R expressed in neuronal cells; indeed, the question of GLP-1R expression in glial cells is still debated. GLP-1R expression in DVC astrocytes was not detected using snRNA sequencing methods [34,97,98]. Nonetheless, it is important to note that while snRNA sequencing is effective at detecting the presence of transcripts, it is not designed to demonstrate the absence of a transcript [98]. Accordingly, Timper et al. [99] recently convincingly demonstrated GLP-1R expression in NTS astrocytes using GFAP IHC coupled to the RNA-Scope detection of GLP-1R mRNA or capture of the fluorescent GLP-1R Liraglutide^594^ in wild-type (WT) mice and negative control GLP-1R knockout (KO) mice [99]. Moreover, the authors also detected GLP-1R expression in astrocytes from the PVN, hippocampus, and ARC. Timper et al. [99] mainly focused their study on ARC astrocytes. The quantification of liraglutide^594^-labelled GFAP astrocytes in the ARC amounted to 35%. A rich combination of complementary approaches on cultured hypothalamic astrocytes elegantly showed that the activation of GLP-1R in glucose-starved astrocytes reduces glucose uptake and promotes b-oxidation. By contrast, GLP-1R KO led to an alteration in mitochondrial oxidative phosphorylation with a compensatory increase in glucose uptake and triggered a cellular stress response characterized by increased expression of fibroblast growth factor 21 (FGF21). The same authors phenotyped a new mouse line through the post-natal pan-astroglial deletion of GLP-1R. Although the mice displayed normal energy homeostasis, they displayed improved systemic glucose homeostasis, which could be reduced through the pan-astroglial deletion of FGF21. The cellular stress response observed in primary cultured astrocytes upon GLP-1R deletion therefore leads to FGF21-induced improvements in systemic glucose homeostasis in vivo. Overall, these results highlight that endogenous GLP-1R signaling in astrocytes is involved in the preservation of mitochondrial metabolic flexibility. In their study, Timper et al. [99] did not assay semaglutide-induced anorexia in mice deprived of GLP-1R expression in astrocytes. Further work is needed to explore the role played by astrocyte GLP-1R expression in the anorectic effects of GLP-1R agonist treatments.

A few years ago, we reported the glial expression of leptin receptor (LepR) isoforms in the adult DVC [19]. In addition to the labelling of NTS and DMNX neuronal cell bodies as described in previous research [100], we observed LepR labelling in DVC glial cells, including vagliocytes. Moreover, using glial primary cell cultures, we found that all the transcripts encoding membrane-bound LepR isoforms were expressed by glial cells originating from the DVC [22]. Notably, the glial expression of LepR in the hypothalamus was reported in the same period [101,102]. The same group also confirmed the functionality of hypothalamic glial LepR by showing the induction of calcium signaling in response to the hormone application using primary astrocyte culture [101]. More recently, Stein et al. [103] compared the reactivity of rat hypothalamic and DVC glia to HFD in terms of leptin responsiveness and astrogliosis. As a first step, they unequivocally demonstrated LepR expression by DVC astrocytes and neurons using RNAscope and detected fluorescent Cy5-Leptin capture by DVC GFAP+ cells located in the *funiculus separens*. Next, they performed calcium imaging on ex vivo brainstem slices to record leptin-triggered calcium signaling in DVC astrocytes and neurons. They showed that about half of the DVC neurons and astrocytes from normal chow-fed rats were responsive to leptin. In addition, HFD strongly reduced this proportion and lessened the amplitude of calcium response in neurons without affecting the proportion in astrocytes. They also showed that the anorexigenic response to icv leptin treatment was reduced by FC pre-treatment in normal chow-fed rats, but that this effect was blunted after HFD exposure for two weeks. Finally, they quantified the GFAP immunoreactivities in the DVC and ARC as a measure of astrogliosis. In WT rats, 8 weeks of exposure to a HFD significantly reduced the DVC GFAP coverage, without affecting that of ARC. Compared to WT rats, obese and diabetic Zucker rats, which were devoid of LepR, exhibited increased astrogliosis in DVC and, by contrast, a sharp reduction in ARC GFAP coverage [103].

Insulin is one of the main humoral signals that modulate energy balance. The deletion of the insulin receptor in the brain leads to increased body fat and insulin resistance [104]. The hypothalamus is one of the major areas targeted by insulin. Thus, injections of insulin into the third ventricle or the mediobasal hypothalamus parenchyma reduced hepatic glucose production [105,106]. Moreover, selective downregulation of insulin receptor expression within the hypothalamus resulted in hyperphagia and increased fat mass [107]. Likewise, the DVC relays the central action of insulin. Insulin reduces hepatic glucose production and food intake at this level [108,109]. Insulin resistance is a hallmark feature of type 2 diabetes and obesity. Of course, insulin resistance affects peripheral organs such as the liver, muscle, and adipose tissue, but insulin resistance also occurs in the brain and contributes to type 2 diabetes and obesity [110]. The development of insulin resistance in the brain was associated with increased inflammation [111] and mitochondrial dysfunction [112]. Remarkably, mitochondria have been reported to change their morphology in response to energy demands [113]. More precisely, mitochondrial fusion occurs in response to high cellular energy demand in order to produce more ATP. During energy excess, however, there is a reduction in mitochondrial activity due to increased mitochondrial fission [114]. Among the proteins regulating mitochondrial dynamics, dynamin-related protein 1 (Drp1) is of particular interest [115]. In fact, Drp1 regulates mitochondrial fission, and increased Drp1 activity results in mitochondrial fission in skeletal muscle [113]. Interestingly, these adaptations are associated with HFD-induced obesity and insulin resistance [113]. In addition, the deletion of Drp1 in the liver prevents HFD-induced insulin resistance in mice [116]. Mitochondrial dynamics seem to play a pivotal role in hypothalamic neurons as well, and the knockdown of proteins involved in the regulation of mitochondrial dynamics in hypothalamic neuronal populations can alter feeding behavior and glucose metabolism [112,117,118,119]. Recently, Filippi et al. [120] reported the HFD-induced alteration of Dpr1 expression in the DVC. Moreover, using an adenovirus-mediated expression of a constitutively inactive form of Drp1, the same group elegantly demonstrated that the inhibition of Drp1 in the DVC protects against developing HFD-dependent insulin resistance and decreases body weight and food intake [121]. Conversely, the DVC expression of a constitutively active form of Drp1 in healthy mice was sufficient to induce insulin resistance, hyperphagia, and body weight gain [121]. These results raise a major question: which DVC cell type is responsible for insulin sensing and resistance? In this context, Patel eta l. [121] demonstrated that the specific inhibition of mitochondrial fission in GFAP-expressing-DVC astrocytes is sufficient to reduce food intake and body weight gain in HFD-fed mice. Interestingly, they also reported that the inhibition of mitochondrial fission in GFAP-expressing-DVC astrocytes prevents the development of HFD-dependent insulin resistance. Nevertheless, how these changes in mitochondrial fission in DVC astrocytes impact insulin sensitivity and neuronal responses remains an open question.

The data regarding the putative neuroactive substances likely to be released from DVC glial cells, which could in turn modulate food intake, are very limited. Nonetheless, it can be reasonably envisioned that DVC glial cells are competent at releasing neuroactive substances classically described as being released by glial cells located in other CNS structures, such as glutamate, ATP, or D-serine [122]. As previously mentioned, ATP release by NTS astrocytes upon ST stimulation and its subsequent action on presynaptic terminals has been reported [78]. Some of these gliotransmitters are known to modulate energy balance when centrally administrated [123,124,125]. However, the contribution of this liberation of gliotransmitters by DVC glial cells to energy balance has not been totally demonstrated. How metabolic signals modulate the putative release of these compounds remains an open question. A recent study showing that the inhibition of connexin 43 hemichannels (Cx43 HCs) decreased food intake could indirectly support the hypothesis of glial release of such neuroactive substances. Indeed, in the CNS, glia, particularly astrocytes, express different Cxs [126]. These Cxs constitute gap junctions (GJ) that contribute to cytoplasmic continuity, providing the structural basis for an extensive astroglial network [127]. Alternatively, connexins not engaged in GJ form hemichannels (HCs), which are permeable by small molecules, such as glucose, ATP, D-serine, and glutamate [128]. Guillebaud et al. [129] reported that Cx43 is strongly expressed by DVC glial cells. Cx43 was often found in the vicinity of synaptic elements. Moreover, the icv injection of TAT-GAP19, a specific blocker of Cx43 HCs, decreases food intake and induces neuronal activation in the hypothalamus and DVC [129]. Altogether, these results suggest a possible release, within the DVC and through glial Cx43 HCs, of molecules with modulatory action on energy balance.

It is not excluded that other molecules of glial origin may contribute to the modulation of energy balance. In this case, endozepines appear to be good candidates. Endozepines constitute a family of peptides that comprises diazepam-binding inhibitor/acyl-CoA-binding protein (DBI/ACBP) and its processing fragments, triakontatetraneuropeptide (TTN) and octadecaneuropeptide (ODN). DBI/ACBP is secreted through an unconventional pathway involving Golgi associated proteins, autophagy genes, and target-soluble N-Ethylmaleimide-Sensitive Factor (NSF) attachment protein receptors (t-SNARES). Endozepines are well known endogenous ligands of benzodiazepine (BZ) receptors (see [130] for review). Over the last two decades, evidence has accumulated that strongly suggests that endozepines could act as endogenous modulators of energy balance by acting at the hypothalamic level [42,130]. The icv injection of ODN dose-dependently reduced food intake in fasted and ad libitum-fed rodents [131,132,133,134,135]. In the CNS, DBI/ACBP is almost exclusively expressed by astrocytes, ependymocytes, and tanycytes [130,133,136]. Interestingly, we recently reported strong glial ODN expression within the brainstem, i.e., the *funiculus separens* and NTS [137]. Further, the injection of ODN C-terminal octapeptide (OP) in the fourth ventricle induced the specific activation of DVC neurocircuitries and reduced food intake [137]. Moreover, reinforcing the idea that endozepines act at the brainstem level in addition to the hypothalamus, the swallowing reflex, which constitutes the first motor component of the ingestion process, triggered by the pre-motoneurons of the swallowing central pattern generator (SwCPG) located in the NTS, is transiently inhibited by the microinjection of OP within the SwCPG [137].

## 7. Open Questions and Future Avenues

As discussed above, there is now evidence supporting the assumption that the DVC glial cells are involved in the modulation of energy balance in both physiological and pathophysiological conditions. However, we believe that this field is still in its infancy and that numerous questions remain open.

One intriguing question is to what extent astrocytes participate in the modulation of glutamatergic signaling within the DVC in the physiological and physiopathological modulation of energy balance. As mentioned above, vagal information is mainly relayed by glutamatergic synapses and glutamate receptors on NTS neurons. Furthermore, astrocytes are widely acknowledged as inseparable components of glutamatergic synapses, and the role of glutamatergic astrocyte-neuron interactions in brain signaling is well documented [138]. The blockade of NTS astrocytic glutamate transporters was previously reported to induce NTS neuron depolarization and reduce the amplitude of vagal afferent-driven excitatory postsynaptic currents with aftereffects on cardiorespiratory function [82,139,140]. In addition to the clearance of extracellular glutamate via glial glutamate transporters, astrocytes can express glutamate receptors, particularly metabotropic glutamate receptor (mGluR) types, whose activation can recruit a variety of molecular cascades, leading to the astrocyte-driven modulation of extracellular levels of glutamate, the release of glutamate and D-Serine, and the activity of neuronal glutamate receptors [138]. While the role of astrocytic AMPAR in the gating of viscerosensory vagal afferent gating has been established, the role(s) of astrocytic mGluR in other synaptic contexts within the DVC remains unknown. At this stage, it is not known whether an alteration in glial glutamate transporters and/or receptors and glutamate release takes place during the short- and long-term modulation of food intake and energy balance. Interestingly, short-term neuroplasticity has been reported within NTS-DMNX synapses and is mechanistically linked to early homeostatic adaptation to HFD [141,142]. More precisely, upon introduction to HFD exposure, rodents display an initial phase of overeating followed by a reduced food intake after 3–5 days to restore caloric balance. This homeostatic regulation is associated with the increased activation of synaptic NMDA receptors at the NTS–DMNX synapse in gastric-projecting DMNX neurons, leading to a subsequent increase in DMNX neuronal excitability and a vagally-mediated increase in gastric motility [141]. More recently, it was shown that the acute HFD-induced increase in synaptic NMDA receptors in the NTS-DMNX synapse is dependent upon the activation of extrasynaptic NMDA receptors. Moreover, the chronic blockade of brainstem extrasynaptic NMDA receptors blunted the HFD-induced delay in gastric emptying and significantly attenuated the homeostatic adaptation to HFD exposure for 3–5 days. Importantly, in the same study, it was shown that HFD maximally activated extrasynaptic NMDA receptors and that this effect could be mimicked by EAAT2 blockade in normal chow-fed rats [142]. Although the role played by DVC glial cells in short-term homeostatic adaptation to HFD has not yet been assessed, the mechanism revealed by Clyburn et al. [142] at the DMNX glutamatergic synapses evidently suggests a decrease in astrocyte-mediated glutamate clearance as a plausible candidate to be explored in the near future. Further investigation into the possible involvement of gliotransmitters in this process is also warranted. In the hypothalamus, HFD-induced astrogliosis was shown to occur in the ARC in a biphasic manner, with an early transient phase that peaks within the first week of HFD and resolves between 2 and 3 weeks [143], followed by a late reinstatement in obese animals that previously consumed HFD for several months [143,144]. In the DVC, the time course of HFD-induced astroglial morphological changes needs to be further explored to determine whether they participate in the initial HFD-induced hyperphagia, the early homeostatic adaptation to preserve energy homeostasis, and later alterations leading to obesity. As mentioned above, mice exposed to HFD for only 12 h displayed hyperphagia that was correlated with an increase in astrocyte number and morphological complexity within the NTS [30]. In the same study, the chemogenetic activation of NTS astrocytes decreased food intake and led to an increase in astrocyte morphological complexity in normal chow-fed rats. These results suggest that NTS astrocytes rapidly react to HFD exposure in a way that allows the limitation of hyperphagia. How this astroglial morphological plasticity evolves over the following 2–5 days of HFD exposure has not been tested, and it remains unclear whether it could account, at least in part, for the concomitant homeostatic adaptation to high-fat diet exposure. Within the same time-frame, the role played by astroglial morphological plasticity in the modulation of NTS-DMNX glutamatergic signaling reported by Clyburn et al. [141] warrants further investigation. Furthermore, in the hypothalamus, long-term HFD exposure leads to the reinstatement of astrogliosis [143,144]. Consequently, two questions may be asked: Is astrogliosis also present in the DVC of obese animals? If so, could this long-lasting astrogliosis modulate glutamatergic transmission in the NTS over the long term and contribute to energy intake dysregulation? Several studies have highlighted changes in vagal afferent responsiveness and NTS neurocircuitries following diet-induced obesity (see [17] for review).

Within the NTS, the decrease in vagal afferent sensitivity and responsiveness results in a decrease in vagal afferent glutamate release. As a result, HFD was shown to be associated with decreased vagal afferent-dependent activation of presynaptic mGluR on inhibitory GABAergic NTS neurons, increasing the inhibitory drive to vagal efferent motoneurons. Following long-term HFD [17], DMNX neurons are also less excitable and less responsive to satiety-inducing neuropeptides, such as CCK and GLP-1, following long-term HFD [17]. Future studies will need to determine whether astrocytes contribute to the reduction of NTS glutamatergic transmission during obesity.

Tanycytes are hypothalamic radial glial-like cells characterized by long processes that contact blood vessels and neurons located within the ventromedial hypothalamus and ARC. Their cell bodies are located on the walls of the third ventricle in contact with the cerebral-spinal fluid (CSF). These anatomical characteristics make these cells possible relays for metabolic signals, allowing them to detect and respond to these different nutritional molecules while being able to functionally couple with neurons in the hypothalamic region. Accordingly, it was reported that tanycytes can respond to different nutritional signals, such as glucose, fatty acids, amino acids, and vitamins (see [145] for review). For instance, in vitro and in situ studies demonstrated that tanycytes sense and respond to extracellular glucose via a rapid Ca^2+^ response that depends on intracellular stores. This response can then propagate to adjacent tanycytes through Ca^2+^ waves dependent on ATP release through Cx43 HCs [146]. Furthermore, the selective destruction of tanycytes through the icv injection of alloxan, a toxin that enters cells through GLUT2, inhibits the counter-regulatory responses generated by hypoglycemia without damaging hypothalamic neurons, which supports the involvement of tanycytes in glucose-sensing mechanisms [147]. Given the strong anatomical and morphological similarities between tanycytes and vagliocytes of the fourth ventricle, it is tempting to imagine that vagliocytes could also be sensitive to nutritional signals and, in turn, participate in the integration of this metabolic information at the brainstem level. This point can be quickly assessed in the future. The recent confirmation of LepR expression in GFAP+ cells located in the *funiculus separens* [103] is a step forward to be followed.

Some authors have reported microglial cell activation within the NTS in response to HFD consumption. Currently, the role of this microglial activation in the mechanisms linked to weight gain following HFD consumption remains poorly characterized and understood. It will undoubtedly be studied in the near future. Another interesting point to address is the possible interactions between astrocytes and microglial cells and their possible contribution to the regulation of the energy balance at the DVC level. Such a dialogue between astrocytes and microglia seems to occur within the AP in the context of the response to inflammation. Indeed, the ip or icv administration of lipopolysaccharides (LPS) induces nuclear translocation of signal transducer and activator of transcription factor 3 (STAT3) in AP astrocytes. Interestingly, pretreatment with minocycline attenuates LPS-induced STAT3 activation in astrocytes located within the AP [148]. These results indicate that astrocyte activation depends on the presence of functional microglia. It would be interesting to know whether such interactions also occur in response to signals linked to the regulation of energy balance.

Finally, we propose to end this overview by discussing a cell population that has emerged very recently in the field of energy balance regulation, namely the oligodendrocyte lineage. Several studies have reported data supporting this idea. Adult animals harbor an abundant population of oligodendrocyte precursor cells (OPCs) that express neural/glial antigen 2 (NG2) chondroitin sulfate proteoglycan, and whose main function is to differentiate into myelination-capable mature oligodendrocytes [149]. Interestingly, Djojo et al. [150] reported that the genetic and pharmacological ablation of adult NG2-glia leads to leptin resistance and obesity in mice. They proposed that NG2-glia cells are essential for the maintenance of dendritic arborization of leptin-sensitive ARC neurons. It was also shown that signaling through GPR17, a G-protein-coupled receptor predominantly expressed in the oligodendrocyte lineage, regulates food intake by modulating hypothalamic neuronal activities [151]. GPR17-deficient mice and mice with an oligodendrocyte-specific KO of GPR17 are resistant to HFD-induced obesity. The invalidation of the *Gpr17* gene increases oligodendrocytic lactate production, which in turn increases POMC activity and reduces food intake. Moreover, Kohnke et al. [152] found that overnight fasting and one-hour refeeding modulated ME oligodendrocyte proliferation and differentiation. In genetically obese ob/ob mice, the nutritional regulations of ME oligodendrocyte differentiation are blunted. This nutritionally induced oligodendrocyte plasticity in turn affects the ME perineuronal networks (PNNs), which are specialized extracellular matrix structures responsible for synaptic stabilization in the adult brain. PNNS are emerging regulators of hypothalamic metabolic functions [153]. To date, no functional data have established the possible involvement of oligodendrocytic lineage cells in the regulation of the energy balance at the level of the DVC. Nevertheless, a few clues allow it to be reasonably considered and at least to be studied in the future. First, Levine et al. [154] reported a slightly higher density of NG2 immunopositive cells and processes in the DVC compared to immediately adjacent regions of the brainstem. Moreover, they demonstrated the ability of these OPCs to react to neuropathological insults in response to viral infection of the brainstem. More recently, two studies using snRNA sequencing to provide a detailed survey of cells within the AP and NTS confirmed the presence within the DVC of both mature oligodendrocytes and OPCs [34,98]. Notably, Dowsett et al. (2021) reported that a large proportion of oligodendrocytes express the glucose-dependent insulinotropic polypeptide receptor (GIPR), whose signaling pathway has been reported to modulate body weight in experimental animals [155]. Moreover, DVC oligodendrocytes were shown to be the most transcriptionally responsive to an overnight fast among DVC neural cells [98]. Based on modified genes and bioinformatics pathway analysis, the authors propose possible DVC oligodendrocyte remodeling in response to an overnight fast, as previously observed in oligodendrocytes in the ME of the hypothalamus [152].

Overall, growing preclinical evidence pointing to the role of DVC glia in energy balance regulation and the development of obesity. Although plausible, no direct evidence for such a mechanism in the development of human obesity and diabetes has been found yet. When compared to the hypothalamus, there is a lack of data at the DVC level. Indeed, there are a few functional magnetic resonance imaging (fMRI) studies that support a role for hypothalamic gliosis in the progression of insulin resistance in obesity and T2D pathogenesis in humans [143,156,157,158]. To conclude, the DVC is a frequently underappreciated and understudied area of the brain that integrates peripheral cues from metabolic status and relays them to the forebrain to control and maintain energy balance. In light of recently published data, it has become evident that glial cells of the DVC must be taken into account in order to better understand the mechanisms regulating energy balance and to develop truly effective strategies against obesity and associated metabolic disorders.

## Figures and Tables

**Figure 1 ijms-23-00960-f001:**
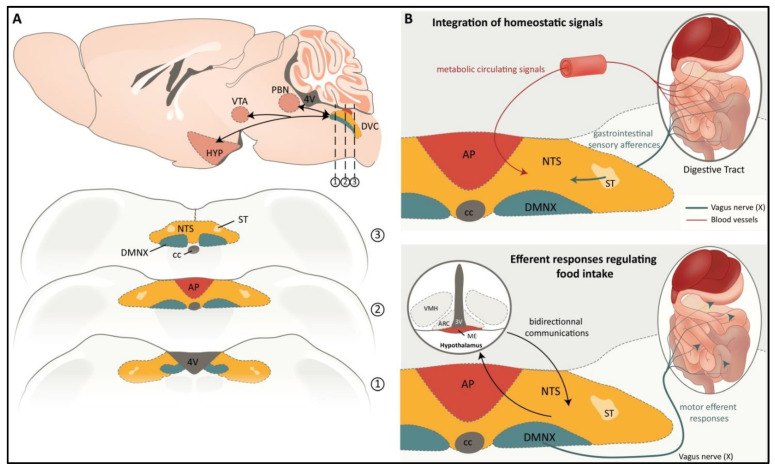
(**A**). A schematic illustration of DVC in the rodent brain, exhibiting its anatomic and functional architecture. The DVC is a brainstem structure located in the vicinity of the 4th ventricle and connected to several centers in the midbrain and forebrain (upper panel). The lower panel illustrates three DVC anatomic levels on coronal sections, i.e., rostral (1), postremal (2), and caudal (3) NTS. (**B**). The NTS integrates gastrointestinal information conveyed by both the vagal sensory afferents and the blood stream (upper panel). Bidirectional communication between the NTS and midbrain and forebrain nuclei, as well as vagal efferents to the gastrointestinal tract, are involved in efferent responses that regulate food intake and energy balance (Lower panel). 3V: 3rd ventricle, 4V: 4th ventricle, ARC: arcuate nucleus, AP: area postrema, cc: central canal, DMNX: dorsal motor nucleus of the vagus nerve, DVC: dorsal vagal complex, ME: median eminence, HYP: hypothalamus, NTS: nucleus of solitary tract, PBN: parabrachial nucleus, ST: solitary tract, VMH: ventromedial hypothalamus, VTA: ventral tegmental areas.

**Figure 2 ijms-23-00960-f002:**
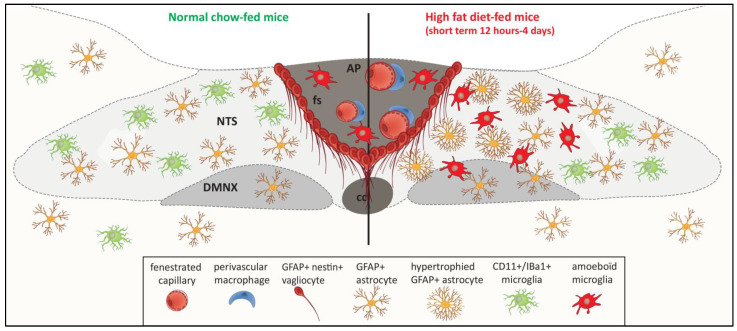
A cartoon illustrating the diversity of glial populations present in the DVC. GFAP+ astrocytes are particularly abundant in the NTS and DMNX when compared to other brainstem nuclei. By contrast, AP is virtually devoid of GFAP+ astrocytes. A population of GFAP+/nestin+ radiating cells, called vagliocytes, is present at the *funiculus separens*, the interface between AP and NTS. In addition, CD11+/Iba+ microglia are present throughout the rostro-caudal axis of the NTS and DMNX [31]. The AP contains a subpopulation of amoeboid CD86+ microglia [43]. An increase in the number and arborization of GFAP+ astrocytes, especially in NTS areas adjacent to AP, shortly after high-fat diet (HFD) consumption has been reported [30]. Moreover, an increase in the number of amoeboid microglia has also been observed in the NTS after HFD consumption (4 days; [44]). AP: area postrema, cc: central canal, DMNX: dorsal motor nucleus of the vagus nerve, fs: *funiculus separens,* NTS: nucleus of solitary tract, GFAP: glial fibrillary acidic protein, CD11: cluster of differentiation 11, Iba1: ionized calcium-binding adaptor molecule 1.

**Figure 3 ijms-23-00960-f003:**
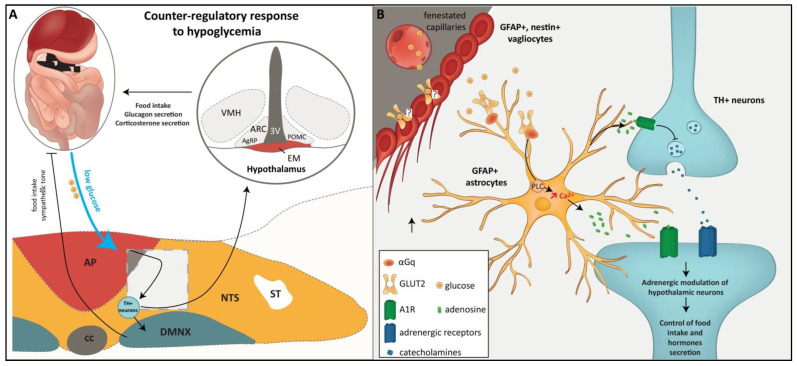
(**A**): A diagram summarizing the involvement of DVC in hypoglycemia detection and subsequent counter-regulatory responses. The NTS contributes to the central detection of glucose availability. In response to hypoglycemia, NTS tyrosine hydroxylase (TH)-positive catecholaminergic neurons elicit a counter-regulatory response that includes glucagon and corticosterone secretion and increases food intake and sympathetic tone [54]. The square delimited by a dotted line indicates the area where panel B originates. (**B**): Schematic representation of glial involvement in DVC sensitivity to hypoglycemia. NTS astrocytes arbor GLUT-2 transporters, whose expression was shown to be necessary for glucagon secretion, neuronal activation, and increased food intake observed in response to glucose deficit. In contrast to tanycytes, the expression of GLUT-2 in vagliocytes has not been reported yet. Glucoprivation induces an intracellular calcium increase in NTS astrocytes. This astrocytic calcium response to glucoprivation is blunted by the blockade of GLUT-2. In turn, NTS astrocyte activation may result in the stimulation, via adenosine release, of TH+ neurons and downstream neuronal circuits responsible for the counter-regulatory response. 3V: 3rd ventricle, AgRP: agouti-related peptide, ARC: arcurate nucleus, AP: area postrema, cc: central canal, DMNX: dorsal motor nucleus of the vagus nerve, PLC: phospholipase C, POMC: pro-opiomelanocortin, ME: median eminence, NTS: nucleus of solitary tract, ST: solitary tract, VMH: ventromedial hypothalamus.

**Figure 4 ijms-23-00960-f004:**
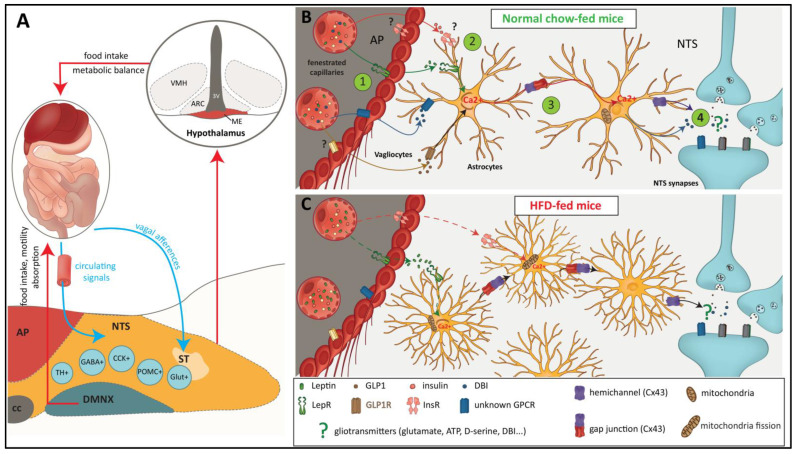
(**A**): Simplified representation of the role of DVC in the regulation of food intake, gastrointestinal functions, and energy balance. Different neuronal populations of the NTS/AP integrate peripheral metabolic signals conveyed by both the vagus nerve and the blood (blue arrows). Then, efferent signals are sent to the hypothalamus and the gastrointestinal tract to regulate metabolic balance (red arrows). CCK: cholecystokinin, GABA: gamma-aminobutyric acid, Glut: glutamate POMC: pro-opiomelanocortin, TH: tyrosine hydroxylase. 3V: 3rd ventricle, AP: area postrema, ARC: arcuate nucleus, cc: central canal, DMNX: dorsal motor nucleus of the vagus nerve, POMC: pro-opiomelanocortin, ME: median eminence, NTS: nucleus of solitary tract, ST: solitary tract, VMH: ventromedial hypothalamus. (**B**,**C**): The role of DVC glial cells in the integration of metabolic signals and the control of food intake is depicted in this diagram. Vagliocytes can help to regulate the diffusion of circulating signals arising from fenestrated capillaries of the AP (bubble 1). NTS astrocytes located in the vicinity of the AP can integrate these metabolic signals (bubble 2). While the diffusion of signaling molecules through astrocytic syncytium is highly probable, thanks to the strong expression of CX43 (bubble 3), the nature of gliotransmitters prone to release by activated astrocytes remains largely unknown (bubble 4). After HFD consumption, the remodeling of glial coverage observed in this area can modify the nature of neuroglia interactions and contribute to the modification of food intake and energy balance.

## Data Availability

Not applicable.

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
