# Peer review of "Glial Modulation of Energy Balance: The Dorsal Vagal Complex Is No Exception"

_ijms, 2022, doi:10.3390/ijms23020960_

Round 1

Reviewer 1 Report

In this manuscript, Troadec et al. provide a rather complete and interesting review of the current knowledge regarding the role of glial cells in the brainstem in the regulation of energy balance, with a particular focus on the DVC, to a point where one can wonder if the title should not be changed and “brainstem” replaced by “DVC”. Beyond this point, as mentioned, the manuscript present a fairly thorough and up-to-date overview of the available literature, allowing to bring some well-deserved light on the still relatively new field of DVC glia in the control of whole-body energy homeostasis, which can only be beneficial for the general understanding of how energy balance is (centrally) controlled.

Nevertheless, if there is one major critique I could formulate, it would be that generally, the quality of English, although not terrible, needs to be improved. There is a substantial amount of mistakes (syntax, spelling, word for word translations..) and I would recommend the authors to have their manuscript to be reviewed by a native speaker. This is especially true regarding the abstract and the introduction. This, overall, harms the manuscript, in my opinion.

In that regard, I have noted a few things, that could serve as examples:

  • line 10: I believe one cannot write “Avoiding overweight”, as overweight is an adjective.
  • lines 14-15: “has been little investigated”: not correct in English.
  • line 46: “many works”: I don’t think “works” work so well here, it does not sound very “native”.
  • lines 162-163: “In accordance, using immunodetection GLT-1 at the light and electron microscope levels…” -> needs rewriting
  • line 173: “Indeed, mice fed with high-fat diet…” -> needs rewriting
  • lines 485-487: needs rewriting

Along the same kind of line, there were also some typos I noted:

  • line 2: Glial ModulaTion of energy balance: The Brainstem Is No Exception
  • line 42: missing a comma after “DVC”
  • lines 46-87: “cholecystokinin” -> font size? baseline?
  • line 141: “arcurate”
  • line 184: “vimentine”
  • line 282: “have been shown to respond to” -> font size? baseline?
  • line 343: “neurons” or “cells missing after “catecholaminergic”
  • line 363: “storeS”
  • line 552: there is an an extra “l” to mitochondria.
  • line 552: missing “have” to go with “undergone” or change the tense.
  • line 565: adenovirus-mediated.
  • line 663: missing comma after “exposure”
  • line 649: “accounts”
  • line 759: “collueagues”

As for the figures, I would recommend improving the legends. I would mention the meaning of all acronyms in any case. Also, some words seem to be removed for a more telegraphic style, which I don't think necessarily works better here (e.g., removing almost all "the").

I also have some other comments below:

  • page 8, figure 3: although regarding the contents of the review justifies focusing on GLUT-2 expression on astrocytes in the brainstem. In a more general setting, GLUT-1 is not known be to mostly expressed by neutrons actually, rather in the BBB or by astrocytes, to my knowledge, which is not really what’s depicted in the schematics here. In addition, the role of GLUT-1 in neutrons is not even discussed.
  • figure 4, panel C: inflammatory molecules acting as modulators released by glial cells were not really discussed within the main text, nor in the figure legend for that matter.

Author Response

We would like to thank the reviewer for the time spent evaluating our manuscript and for his (her) positive and constructive feedback. The manuscript has been corrected, taking into account most of the criticisms and suggestions raised. The text has been edited and corrected by a native English speaker. The title and figures have been modified as suggested.

Reviewer 2 Report

This review presents comprehensively up-to-date information about Glial modulation of energy balance using primary data obtained in rodents. The illustrations are excellent. The text follows a well-designed plan with very welcome paragraphs like paragraph 7 - open questions and future avenues. 

-The text is nevertheless challenging to follow and will be significantly improved after in-depth correction by an English native. For instance, the first three pages could be easily shortened by removing the relative-containing sentences. 

  • Some sections required a more detailed description. For instance, lines 145 to 165 refer to indefinite adjectives such as ‘weak,’ ‘typical,’ ‘relative abundance’ while a receptor density could be either increase or decrease. Surprisingly, line 470 and follow this problem is not present anymore when dealing with GLP-1r. Some laboratory slang is also present in paragraph 3.
  • Inaccurate use of the 'Brain' instead of the hypothalamus. Line 549, the authors refer to ‘brain’ for reference 110 while insulin resistance is only present in the hypothalamus.
  • Translational aspect - Comments on the possible (likely ?) difference between the vast amount of data coming from rodents and their translation to human and medicine.

Author Response

We would like to thank the reviewer for the time spent evaluating our manuscript and for his or her positive and constructive feedback. The manuscript has been corrected, taking into account most of the criticisms and suggestions raised. The text has been edited and corrected by a native English speaker. A paragraph dealing with the translational aspects has been added at the end of paragraph 7.